

# Joint classification and regression with deep multi task learning model using conventional based patch extraction for brain disease diagnosis

Padmapriya K. and Ezhumalai Periyathambi

Department of Computer Science and Engineering, RMD Engineering College, Chennai, Tamil Nadu, India

## ABSTRACT

**Background:** The best possible treatment planning and patient care depend on the precise diagnosis of brain diseases made with medical imaging information. Magnetic resonance imaging (MRI) is increasingly used in clinical score prediction and computer-aided brain disease (BD) diagnosis due to its outstanding correlation. Most modern collaborative learning methods require manually created feature representations for MR images. We present an effective iterative method and rigorously show its convergence, as the suggested goal is a non-smooth optimization problem that is challenging to tackle in general. In particular, we extract many image patches surrounding these landmarks by using data to recognize discriminative anatomical characteristics in MR images. Our experimental results, which demonstrated significant increases in key performance metrics with 500 data such as specificity of 94.18%, sensitivity of 93.19%, accuracy of 96.97%, F1-score of 94.18%, RMSE of 22.76%, and execution time of 4.875 ms demonstrated the efficiency of the proposed method, Deep Multi-Task Convolutional Neural Network (DMTCNN).

**Methods:** In this research present a DMTCNN for combined regression and classification. The proposed DMTCNN model aims to predict both the presence of brain diseases and quantitative disease-related measures like tumor volume or disease severity. Through cooperative learning of several tasks, the model might make greater use of shared information and improve overall performance. For pre-processing system uses an edge detector, which is canny edge detector. The proposed model learns many tasks concurrently, such as categorizing different brain diseases or anomalies, by extracting features from image patches using convolutional neural networks (CNNs). Using common representations across tasks, the multi-task learning (MTL) method enhances model generalization and diagnostic accuracy even in the absence of sufficient labeled data.

**Results:** One of our unique discoveries is that, using our datasets, we verified that our proposed algorithm, DMTCNN, could appropriately categorize dissimilar brain disorders. Particularly, the proposed DMTCNN model achieves better than state-of-the-art techniques in precisely identifying brain diseases.

Corresponding author
Padmapriya K.,
padmapriyakresearch@outlook.com

## INTRODUCTION

The human brain, which controls numerous essential processes in the body, is the height of complexity and importance. Among its many functions, it stimulates creativity, deliberates on issues, makes decisions, and manages memory and information storage and retrieval. The entirety of our life's history is preserved in our physical memory, which is crucial in forming our identities and character. Dementia-related memory loss and losing our sense of surroundings can be terrifying experiences (*Huang et al., 2020*). Alzheimer's disease (AD), which makes up about 60–70% of all dementia cases (*Castellazzi et al., 2020*), is one of the most mutual types of dementia. Data from the World Alzheimer's Survey show that 78 million new cases of Alzheimer's disease will happen by 2030, and an estimated 55 million people will already have the illness. AD not only has terrible effects on the people who have it and the people who care for them, but it also costs a lot of money. It is thought that AD-related costs will hit $345 billion in the USA alone by 2023 (*Huang, Chao & Hu, 2020*). However, this is really just the tip of the iceberg. Our present situation has been called an AD outbreak (*Alzheimer's Association, 2023*), and because the population is skewed toward older people, costs are expected to triple by 2050.

The main distinction between primary and secondary brain tumors is their origin within the brain. Of all brain malignancies, 70% are caused by primary tumors, while the residual 30% are secondary tumors (*Akil, Saouli & Kachouri, 2020*). Main brain tumors invent from brain cells, whereas secondary brain malignancies start in another organ and travel through the bloodstream to the brain. An NBTF study projects that 29,000 new cases of primary brain tumors are diagnosed in the USA each year, with approximately 13,000 of those individuals dying from the tumor.

Meningiomas frequently development from the membranes that surround the brain and spinal cord. These tumors are known for their sluggish improvement and are therefore considered less dangerous. On the contrary, pituitary tumors develop inside the gland that produces a variety of hormones and is located close to the base of the brain. Nevertheless, regularly harmless, they might cause serious importance such as hormone overproduction, visual impairment, or hormonal deficiencies (*AlSaeed & Omar, 2022*). Uppermost detection of pituitary tumors is important and has significant therapeutic implications. If left undetected, these tumors can be deadly or cause lasting disability. Additionally, research on brain function has been notably developed by augmentations in medical imaging, mainly in neuroimaging methods like MRI. Using artificial intelligence to find AD is hard for researchers for a number of reasons. To begin, the quality of medical images isn't always good, and there are mistakes in preparation and brain segmentation. There are things like noise, artifacts, and technical limits that can lower the quality of medical images (*Weiner et al., 2024*), which can make AD detection algorithms less accurate. Also, mistakes in the pre-processing and segmentation techniques make it even harder to analyze these pictures accurately.

Another challenge is that there aren't any complete datasets that include a lot of different people and biomarkers. To develop powerful AD detection models, you must have access to a diverse set of datasets that represent different phases of the illness and

include several indicators (*Alsubaie, Luo & Shaukat, 2024*). Finding datasets with a sufficient number of participants can be challenging, which increases the difficulty of successfully training and testing AI models.

Alzheimer's disease (AD) is an incurable brain illness that affects millions of people worldwide. Early AD diagnosis is critical for improving patients' lives and ensuring they receive the appropriate care and medications. To start the training process, the matched-filter approach is employed to make 3D images more contrasty and less noisy or outlier-filled. The ADNI is used, which includes fMRI data from 675 participants. The fMRI data contains 285 features derived from the robust multitask feature learning approach. The Mini-Mental State Examination (MMSE) score is used to classify the severity of Alzheimer's disease into low, mild, moderate, and severe categories. The sample data for the deep learning model's training job includes 285 features extracted from an fMRI picture as well as the patients' MMSE scores. The training data contained information about 800 cases whose characteristics had been standardized. The test group includes 200 sets of features, each with an MMSE score. Following that, the PCA approach is used to determine which and how many features to include. The results demonstrate that 167 features explain 98% of the variation in all 285 features. To classify the data, a variety of machine learning methods are utilized, including k-neural network (KNN), support vector machine (SVM), decision tree (DT), latent Dirichlet allocation (LDA), and random forest. It becomes found that the KNN, SVM, DT, LDA, RF, and provided CNN algorithms are accurate 77.5%, 85.8%, 91.7%, 79.5%, and 85.1% of the time respectively (*Veerappampalayam Easwaramoorthy et al., 2022*).

This research describes a novel technique to brain illness diagnosis based on DMTCNN. Our description combines patch extraction methods with a multi-task learning framework, allowing for concurrent classification and regression. Our system extracts patches from brain images, capturing the fine-grained characteristics obligatory for effective diagnosis. The DMTCNN model uses these features to perform simultaneous classification. (*e.g.*, illness kind recognition) and regression (*e.g.*, disease progression estimation), subsequent in advanced diagnostic precision and efficiency. We inspect the design and training approaches of the DMTCNN, highlighting its capacity to accomplish shared demonstrations that help both tasks. Through extensive trails on benchmark brain imaging datasets, we display that our systems outperform conventional single-task models. Furthermore, we create that combining classification and regression into a single model decreases computing complexity while improving diagnostic system reliability. Using deep learning to diagnose brain illnesses has advanced provocatively with the enhancement of the DMTCNN-based patch extraction technique. It not only progresses diagnostic accuracy and efficiency but also serves as a scalable framework for future research in multi-task learning and medical imaging.

## Motivation of this research

- Brain imaging data is multidimensional and complex. The hybrid classification and regression method optimize the use of available data, potentially important to enhanced diagnostic insights and more robust models.

- Patch-based extraction approaches can concentrate on crucial areas of brain imaging, improving the feature extraction process. DMTCNN can successfully acquire and utilize these properties to enhance diagnostic performance.
- Effective therapy depends on a timely and precise diagnosis of brain diseases. With the potential to predict sickness progression and identify early signs, the DMTCNN-based approach enables prompt and suitable therapies.
- The regression portion of DMTCNN can provide unique insights into illness severity and progression, allowing for customized treatment approaches for individual patients.

**The main contribution of this research**

- To progress the detection of brain diseases, quantitative disease-related measures like tumor volume or disease severity are predicted by the proposed DMTCNN model.
- Compared to single-task models, the proposed model that addresses both classification and regression tasks captures complex patterns, leading to more accurate and informative diagnoses.
- Targets important regions associated with brain illnesses by minimizing computing overhead and increasing efficiency through the extraction of smaller, targeted patches from brain imaging data.
- Faster diagnosis times are achieved by the integration of multi-task learning and patch extraction, which is crucial in clinical situations where timely decision-making is essential.
- The proposed model learns many tasks concurrently, such as categorizing different brain diseases or anomalies, by extracting features from image patches using CNNs.
- Finally, the suggested DMTCNN model performs better at accurately recognizing brain illnesses than the most advanced methods.

The remainder of our study expands on this structure: The literature evaluation is covered in "Literature Survey", and the proposed research is summarized in "Proposed System". The datasets, evaluation standards, and experimental findings are shown in "Result and Discussion". Finally, "Conclusion" provides a summary of this work's conclusions.

## LITERATURE SURVEY

In addition to emphasizing the most recent diagnostic methods and technologies, this literature review investigates the current status of research in brain disease diagnosis. By analyzing recent studies and breakthroughs, they aim to highlight the positive and negative aspects of current diagnostic approaches, identify emerging trends, and indicate prospective areas for future research. This survey will provide an in-depth overview of the field's progress and challenges, supporting ongoing efforts to improve brain disease diagnosis and patient outcomes.

*Liu et al. (2020)* developed a deep learning framework that uses CNNs to automatically segment the hippocampal region and classify AD. This author used a 3D densely

connected convolutional network (3D DenseNet) to extract 3D patch features for classification using data from hippocampal segmentation. The Alzheimer's Disease Neuroimaging Initiative (ADNI) database provided baseline T1-weighted structural MRI data from 97 AD patients, 233 MCI patients, and 119 NC patients. They evaluated our technique. Hippocampal segmentation with the proposed approach yields 87.0% Dice similarity.

*Pusparani et al. (2023)*, certain MRI imaging viewpoints are more accurate in identifying AD patients. Using the ResNet50 and LeNet designs, they performed multiclass classification on the ADNI dataset (https://adni.loni.usc.edu/). Our examination included three different perspectives and classifications. After applying these viewpoints and classifications to a total of 4,500 MRI slices, our study found that the use of individual slices, as opposed to whole slices, produced better results for AD classification in MRI images. The coronal view results were mainly notable as they recognized performance like to the assessments normally given by medical professionals to diagnose AD.

*Duc et al. (2023)* conducted an automatic diagnosis of AD using a three-dimensional CNN model. The MMSE score for Alzheimer's patients was predicted using tree regression, support vector regression, linear least square regression, group-independent component analysis, and bagging-based ensemble regression. Furthermore, the performance of MMSE regression was enhanced through the application of SVM-based recursive feature reduction. Three main challenges that the model presented here must overcome in order to diagnose Alzheimer's disease are class imbalance, overfitting, and fading gradient.

*Khatri & Kwon (2022)* investigated dynamic frequency functional networks at frequency response time series, which included full-band, slow-4, and slow-5 bands, using the rs-fMRI data from the ADNI. His team merged four frequency bands with dynamic frequency features from the functional networks of the brain to help in the early detection of AD. It also provides early Alzheimer's detection and a novel viewpoint on the functioning of the brain network. Achieved results included 94.10% classification accuracy, 96.75% specificity, and 90.95% sensitivity for the author. By utilizing different degrees of evaluation matrix, the High-Order Dynamic Functional Connectivity model's experimental results can enhance the classification performance for AD detection.

To address these issues, *Sethuraman et al. (2023)* provide a computerized AD diagnosis method. It makes use of several unique deep learning models for tenfold cross-validation and objective assessment. The researcher discovered that AD illnesses can be distinguished from NC by using whole band ranges, slow4 and slow5, or higher and lower frequency band techniques. The first technique treats AD with SVM and KNN. Second, ADNI organizations' rs-fMRI datasets are used with customized AlexNet and Inception blocks. We modified parameters to try alternative machine learning and deep learning methods and achieved good accuracy. Our three-band approach performs well without external feature selection. Our technique distinguishes AD patients from controls with accuracy (96.61%)/area under the curve (AUC) (0.9663).

*El-Assy et al. (2024)* developed a CNN design that uses MRI data from the ADNI dataset. In the classification layer, this network combines two CNN models, each with

different filter widths and pooling layers. Three, four, and five kinds are used to solve the multi-class difficult. These great accuracy rates found the network's ability to extract and distinguish important elements from MRI images to precisely diagnose AD stages and subtypes. Through hierarchical convolutional, pooling, and fully connected layers, the network design extracts local and global patterns from the data to exactly distinguish AD kinds. Accurate AD classification impacts early detection, particular treatment planning, disease monitoring, and prognosis.

Multi-Level FC Fusion (MFC) is a novel categorization technique that *Liang & Xu (2022)* presented with the objective of distinguishing among numerous brain diseases. To classify patterns of functional connectivity (FC) in both high- and low-order domains, the technique begins by increasing an advanced deep neural network (DNN) model that can extract and improve abstract feature demonstrations. Prototype learning is incorporated into the procedure during supervised fine-tuning to develop the separability and compactness of features both within and among classes. The DNN model underwent both supervised and unsupervised learning stages during its training. They developed an ensemble classifier to categorize brain illnesses using hierarchical stacking learning. To develop accuracy, our technique used multi-level abstract FC features. Two large fMRI datasets were systematically tested.

*Khatri & Kwon (2024)* suggested combining convolution-attention processes in transformer-based classifiers for AD brain datasets as a means of enhancing performance without consuming a lot of processing power. Switching from multi-head attention to lightweight multi-head self-attention (LMHSA), utilizing inverted residual (IRU) blocks, and including local feed-forward networks (LFFN) yields the best results. When trained on AD datasets using a gradient-centralized approach, Adam has a high success rate of 94.31% for multiple class classification, 95.37% for binary classification (AD *vs.* HC), and 92.15% for HC *vs.* MCI.

In order to reduce ambiguity in the diagnosis of brain disorders across several centers with respect to center, feature, and label, *Zhu et al. (2023)* introduced a novel method known as the Multi-Discriminator Active Adversarial Network (MDAAN). The first step in mitigating domain shift caused by adversarial learning is to derive latent invariant representations of the source and target centers. The method examines the variations in data distribution between the source and target centers in order to dynamically evaluate the source center's fusion contribution. With minimal sample annotation cost, only target center hard learning samples are labeled. To minimize negative transfer and reinforce the multi-center model, they lastly employ the chosen samples as the auxiliary domain.

## Limitations of existing system

- Large amount of annotated data is needed to train medical DL models for classification and regression. The model's robustness and accuracy are limited by the difficulty of obtaining labeled brain imaging data with diagnostic and severity labels.

- Despite advances in execution time, MTL model training still demands a lot of computer power, especially with big, high-resolution brain imaging datasets. This may limit accessibility in resource-constrained or smaller clinical environment.
- Some diagnostic procedures, such as biopsies, are intrusive and can cause difficulties and discomfort for patients.
- Some diagnostic approaches, particularly those that require laboratory analysis or complex imaging, can be time-consuming, delaying diagnosis and treatment.
- Many systems are not equipped to detect brain disorders at the earliest stages, resulting in inadequate patient outcomes.

## PROPOSED SYSTEM

This section describes the process of extracting several image patches around these landmarks by employing anatomical characteristic recognition data from MR images. After that, we present DMTCNN for combined regression and classification. The presence of brain diseases and quantitative disease-related measures like tumor volume or disease severity are predicted by the proposed DMTCNN model. Through cooperative learning of several tasks, the model can make greater use of shared information and improve overall performance. The proposed model learns many tasks concurrently, such as categorizing different brain diseases or anomalies, by extracting features from image patches using CNNs. Using common representations across tasks, the MTL method enhances model generalization and diagnostic accuracy, even in the absence of sufficient labeled data. Figure 1 shows the DMTCNN block diagram.

### Dataset

We assessed the proposed methodology in this work using two distinct brain MRI datasets. Our primary dataset, the "Figshare brain tumor dataset," was located at (*Cheng, 2017*). It is well known for having one of the biggest collections and for being an essential tool in the diagnosis of brain cancers. This collection includes 3,064 genuine brain MRI samples from 233 people. Of these, the meningioma class includes 708 samples, the pituitary class includes 930 examples, and the gliomas class includes 1,426 samples. Every image has the same dimensions of $512 \times 512$ pixels. In contrast, the Brain MRI Dataset was reduced in size and distributed in *Kaggle (2020)*, *Wang et al. (2024)*. There are 253 MRI samples in all, with a pixel size of 845 by 845. This dataset includes 155 MRI samples with tumors. It is important to remember that the MRI examples show a variety of structural complexity, devices employed, noise levels, bias field effects, and other features. The T1-weighted MRI image collection was chosen because of its contrast-enhanced quality. As a result, it simplifies distinguishing the tumor-affected areas, making them preferred for treatment planning. The proposed DMTCNN approach requires brain MRI scans to diagnose neurological disorders. Weighted according to T1, T2, FLAIR, and Diffusion These photos include imaging MRI sequences. Each sequence displays different brain tissue features, helping identify tumors, atrophy, and lesions. T1 scans provide anatomical details, but T2- and FLAIR images show aberrant brain fluid accumulations, which are frequent in

**Peer**J Computer Science

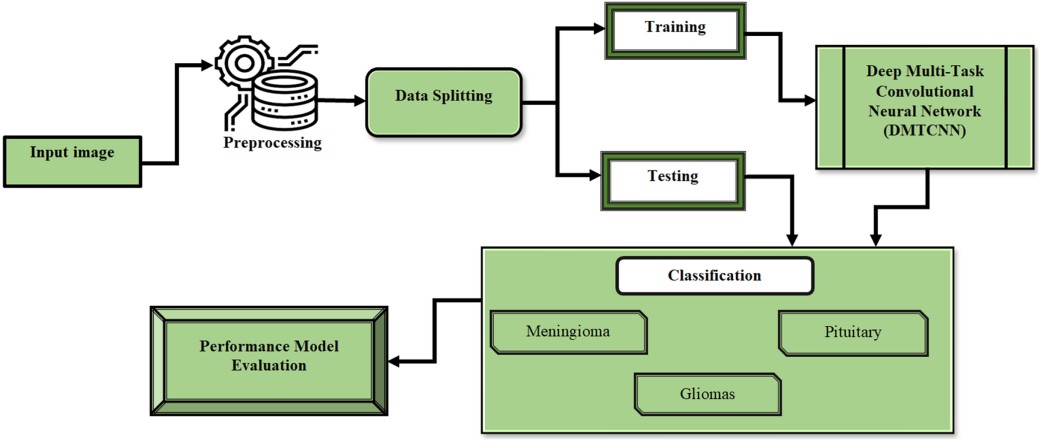

**Figure 1  Block diagram of DMTCNN.**

multiple sclerosis and strokes. The deep learning model can identify disease-relevant features for classification (*e.g.*, detecting Alzheimer's or tumors) and regression (*e.g.*, predicting disease severity or progression) by extracting patches from these MRI types, improving diagnostic accuracy and brain health assessment.

## Pre-processing

The Canny edge detector is a pre-processing approach for detecting edges in images. It is particularly successful in distinguishing significant edges from noise by employing a number of stages such as Gaussian smoothing, gradient calculation, non-maximum suppression, and hysteresis thresholding.

In order to assist in the diagnosis of brain disorders, the Canny edge detector advances the edges of brain pictures, such as MRI or CT scans. This improvement aids in detecting structural anomalies or lesions that may designate diseases such as tumors, hemorrhages, or neurodegenerative disorders. The detector is well-defined by the steps below:

**1. Gaussian smoothing:** Smooths the image to reduce noise and annoying details.

$$Smoothed\_Image(x, y) = (f * G)(x, y) \tag{1}$$

where $f$ is the original image and $G$ is the Gaussian kernel.

**2. Gradient calculation:** Computes the gradient magnitude and orientation to find potential edges.

$$Gradient\_Magnitude(x, y) = \sqrt{G_x(x, y)^2 + G_y(x, y)^2} \tag{2}$$

$$Graident\_Orientation(x, y) = \tan^{-1}\left(\frac{G_y(x, y)}{G_x(x, y)}\right) \tag{3}$$

where $G_x$ and $G_y$ are the derivatives of $f$ in the $x$ and $y$ directions.

**3. Non-maximum suppression:** Thins down the edges to single pixel width to accurately locate edge points.

$$Edge\_Strength(x,y) = \begin{cases} Gradient\_Magnitude(x,y) & if\ the\ pixel\ is\ a\ local\ maxim \\ 0 & otherwise \end{cases} \quad (4)$$

**4. Hysteresis thresholding:** Decides which edge pixels are part of the actual edges by applying two thresholds.

$$Edge\_Map(x,y) = \begin{cases} Strong\_Edge & if\ Edge\_Strength(x,y) \geq High\_Threshold \\ Weak\_Edge & if\ Low\_Threshold \leq Edge\_Strength(x,y) < 1 \\ No\_Edge & Otherwise \end{cases} \quad (5)$$

The first step in the multi-stage process used by the Canny edge detector is Gaussian smoothing, which smooths and reduces noise in the image to increase edge detection accuracy. The gradient of the picture is then computed using Sobel operators to ascertain the direction and size of intensity variations. The identified edges are then thinned *via* non-maximum suppression, leaving only the local maxima, aiding in the creation of a more lucid edge representation. To categorize pixels into strong edges, weak edges, and non-edges, the algorithm uses double thresholding, making sure that only the most important edges are kept. Finally, weak edges are eliminated and continuity is ensured by edge tracking using hysteresis, which links weak edges to strong ones. The Canny edge detector is an essential tool for a wide range of applications, including object recognition, image segmentation, and feature extraction in robotics, autonomous systems, and medical imaging. Its comprehensive approach reduces the likelihood of false edge detection and yields well-defined, thin edges. Because of its precision and robustness, it has become a typical technique for edge detection tasks.

## Deep multi-task convolutional neural network

However, most practical applications do not enable task grouping ahead of time. The feature-based grouping approach is limited to measuring task correlation. When several tasks need to be put together, the situation becomes more difficult. We need a more universal approach. Additionally, in the context of multi-task learning, collaborative feature acquisition is more important than the degree of task sharing. Regarding the output, the objective of supervised learning is to extract and represent the pertinent data from the input variable. A model has a limited capacity to communicate information. Thus, if the jointly learned tasks are highly diverse, the model might become overfitted. As a result, the extent to which activities share information should be variable, particularly in situations where the relationships between the grouped tasks are not entirely evident.

We suggest using the deep multi-task CNN model to address the challenges presented by multi-task learning issues. The primary dissimilarity amongst this approach and the traditional CNN model is the number of deep connections between subnets of various tasks. Within a work group, the lower layers share supervisory signals from the higher layers. Gradient descent modifies the effect of a guiding signal in a specific lower layer to facilitate the learning of task transfer connections (TTC). This method automatically categorizes tasks into nuanced categories during the training phase. As a result, task

linkages are no longer evaluated in a simple binary framework of related or unconnected; rather, they are measured based on their level of significance (*Rohani, Farsi & Mohamadzadeh, 2023*). The suggested model degenerates into numerous single-task learning models when the TTCs within a similar subnet $\alpha^{pq}(p = q)$ are 1 and those across tasks are 0. The proposed model becomes an average multi-task CNN model when the TTC factors $\alpha^{pq}(p, q \in taskset)$ are the same.

The function $f(.)$ of weights $W$ and input $A$ represents the standard CNN model:

$$f(A, W). \tag{6}$$

The purpose is to reduce the discrepancy between the output $B$ of $f(A, W)$ and the ground truth:

$$\min(D) \tag{7}$$

where:

$$D = -[B \ln(F) + (1 - B) \ln(1 - F)]. \tag{8}$$

A function of $A$, $W$, and TTC $\alpha$ is used to express the deep multi-task CNN model in the presence of $n$ tasks:

$$F(A_1, A_2, ...., A_n, W_1, W_2, ...., W_n, \alpha). \tag{9}$$

Additionally, the multi-task model aims to minimize the cost of each task

$$\min \left( \sum_{p=1}^{n} (D_p) \right) \tag{10}$$

where:

$$D_p = -\left[ B_p \ln(F_p) + 1(1 - B_p) \ln(1 - F_p) \right] \tag{11}$$
$$F(A_p, W_1, W_2, ..., W_n, \alpha). \tag{12}$$

As the definition of the TTC factor, $\alpha$ is projected to be between 0 and 1. Consequently, we introduce auxiliary variable $\beta$ where:

$$\alpha = sigmoid(\beta), \beta \in (-\infty, +\infty). \tag{13}$$

Our multi-task model differs from the traditional CNN model primarily in how weights are updated during training. This is because, in our paradigm, every subnet must consider supervisory signals from both other subnets and its own subnet. Several optimization techniques, including adaptive gradient, RMSprop, and stochastic gradient descent, have varying weight updating details. As such, we only display the generic forms. Weight updating in the traditional CNN model can be shown as follows:

$$W_{t+1} = W_t + V_{t+1}. \tag{14}$$

Algorithm 1 is used in our framework to update the weights, providing weights for all $\alpha^{pq}$ components in the TTC, n positions, and $W^1, W^2, \ldots\ldots, W^n$.

The first algorithm uses $\beta_t^{pq}$ as an auxiliary variable, $V_{t+1}^{W^{pq}}$ as the $W_{t+1}^p$ and BPQ update values for a particular optimization algorithm to determine the cost of task $q$, $\alpha_t^p$ as the corresponding $t+1$ factor of $\beta^{pq}$ to task $q$, which regulates the effect of the supervisory signal from task $q$ on task $p$, and $W^p$ and $W_t^p$ as the weights at iteration $t$ and $t+1$, respectively.

$$V_{t+1}^{W^{pq}} = \frac{\partial D_q}{\partial W^q} \cdot \frac{\partial W^q}{\partial W^p} \tag{15}$$

$$V_{t+1}^{\beta^{pq}} = \frac{\partial D_q}{\partial \alpha^{pq}} \cdot \frac{\partial \alpha^{pq}}{\partial \beta^{pq}}. \tag{16}$$

Particularly, supervisory signals from other subnets will be weaker in order to maintain the dominance of information from the subnet itself. Consequently, when $p = q$, $\alpha_t^{pq}$ is set to 1. Batch normalization is used to reduce feature value range volatility.

Our model is superior to the conventional training strategy because it can add additional tasks incrementally while maintaining the completeness of each task branch. As training progresses, the subnets of the newly assigned tasks link to the taught tasks and learn their own parameters in the process. The parameters of the associated subnets are fixed for the tasks that have already been learned. Figure 2 depicts a scenario in which two tasks have been trained and the model has been updated to include a novel task.

Advantages of proposed method

- Sharing representations across tasks is one way that multitask learning might result in more effective learning processes. Due to the network's improved ability to generalize from the shared data, this can lower the risk of overfitting, particularly in situations with a lack of labeled data.
- The patch-based extraction technique may improve the network's capacity to identify minute alterations linked to brain disorders by concentrating on pertinent local brain regions. This specialized strategy may result in the identification of anomalies that are more sensitive and exact.
- The computing cost can be greatly reduced by extracting patches rather than processing complete brain scans, which makes the technology more practical for large-scale or real-time applications. Additionally, it makes it possible to analyze particular brain regions in greater detail and with greater focus. Table 1 shows that description of mathematical symbols used in this research.

## RESULT AND DISCUSSION

This section discusses the experimental setup, followed by a comparison of methodologies, and finally, the performance measures.

---

**Algorithm 1** **Parameter updates.**

Input: Labeled data of task set $T$.

1: Initialize $W_0^p$ and $\alpha_0^{pq}(p, q \in T)$.

2: while not converged do

3:     \\Update the weights of each CNN subnet:

4:     for $p \in T$ do

5:         $W_{t+1}^p = W_t^p + \sum_{p=1}^n \alpha_t^{pq} V_{t+1}^{W^{pq}}$

6:     end for

7:     \\Update the TTC factors:

8:     for $p \in T$ do

9:       for $q \in T$ do

10:         $\beta_{t+1}^{pq} = \beta_t^{pq} + V_{t+1}^{\beta^{pq}}$

11:         $\alpha_{t+1}^{pq} = sigmoid\left(\beta_{t+1}^{pq}\right)$

12:       end for

13:     end for

14: end while

---

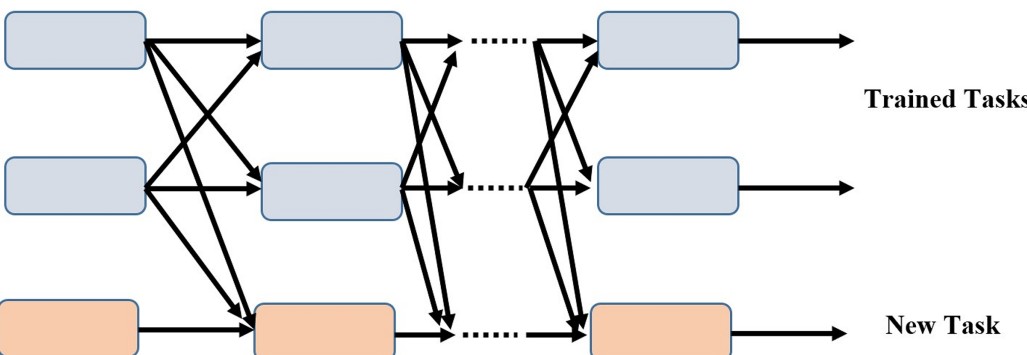

**Figure 2** **Incremental task transfer learning using two trained tasks and one new task.**

---

**Table 1 Description of mathematical symbol.**

| Symbol | Description |
|---|---|
| $f(\cdot)$ | Function |
| $W$ | Weights |
| $A$ | Input |
| $B$ | Output |
| $f(A, W)$ | Ground truth |
| $n$ | Tasks |
| $W_{t+1}^p$ | Updated weights |
| $t + 1$ | Iteration |
| $q$ | Task |

## Experimental setup

The Python development environment utilized for the studies was the Google Colaboratory Pro (*Google Colaboratory, 2019*) platform, also known as Colab Pro. With this cloud service from Google, users can create and run Python programs on a hosted GPU. A ratio that can be used is 70% of the data for training, 15% for validation, and 15% for testing. The model is capable of learning features for both classification and regression tasks from the training set, which contains the majority of the data. This ensures that the DMTCNN model can generalize effectively to new data. During the training phase, the validation set is employed to evaluate the model on unobserved data in order to fine-tune hyperparameters and prevent overfitting. Lastly, the model's performance on both classification and regression tasks in a real-world context is evaluated using the test set, which is entirely isolated from the training process. The model's robust performance when applied to new, unseen cases is critical for reliable brain disease diagnosis, and this balanced approach ensures that it not only learns efficiently but also performs robustly. In Table 2 shows outlines typical parameter values for DMTCNN model using conventional patch extraction for brain disease diagnosis.

## Comparative methods

**Deep neural network (DNN) (*El-Assy et al., 2024*):** Prototype learning was added to the supervised and unsupervised learning stages of the DNN model's fine-tuning procedure in order to improve both intra-class similarity and inter-class differentiation in feature representations.

**Deep convolutional neural network (DCNN) (*Islam & Zhang, 2018*):** A recent approach uses a binary classifier to gain more efficiency in early-stage analysis and to correctly identify different stages of AD.

**Alzheimer's disease diagnosis using enhanced manta ray foraging optimization based deep learning (ADD-EMRFODL) (*Pusparani et al., 2023*):** GF is used in the ADD-EMRFODL method as an initial processing step. Additionally, it generates feature vectors using the DenseNet-121 model.

**Mask region-based convolution neural network mask R-CNN (*He et al., 2017*):** Tumor delineation accuracy is greatly improved by this technique, which uses area-based segmentation to create a mask rather than relying on a threshold or border model for precise segmentation.

## Performance metrics

True positives (TP) are instances in which the classifier accurately detects positive cases. True positives (TP) are situations that have been appropriately identified as positive. False positives (FPs) occur when negative cases are incorrectly labeled as positive. FP stands for false positives. True negatives (TN) are occasions in which the classifier correctly detects negative cases. The count of true negatives is referred to as TN. Positive occurrences that are wrongly labeled as negative are known as false negatives (FN). The number of false negatives is indicated by FN.

**Table 2 Parameter values for DMTCNN model.**

| Parameter | Description | Typical values |
|---|---|---|
| Patch size | Size of the patches extracted from brain imaging data | $32 \times 32$, $64 \times 64$ pixels |
| Learning rate | Step size used for updating model weights during training | 0.001–0.0001 |
| Batch size | Number of samples processed before model update | 16, 32, 64 |
| Epochs | Number of complete passes through the training dataset | 100–500 |
| Activation function | Function applied to model outputs to introduce non-linearity | ReLU, Leaky ReLU |
| Validation split | Proportion of data set aside for validation | 15% |
| Test split | Proportion of data set aside for final testing | 15% |

Accuracy: In contrast to the total number of samples evaluated, it assesses the proportion of correctly classified samples, both positive and negative.

$$ACC = (TP + TN)/(TP + TN + FP + FN). \tag{17}$$

Sensitivity: It evaluates the accuracy with which positive samples are identified.

$$SEN = TP/(TP + FN). \tag{18}$$

Specificity: With regard to the total, it evaluates the capacity to identify negative samples.

$$SPE = TP/(TP + FP). \tag{19}$$

F-measure: The variance is computed by subtracting the total number of false positives from the total number of genuine positives.

$$F - Measure = 2 * \frac{(precision * recall)}{(precision + recall)} \tag{20}$$

### Comparative analysis of performance metrics

Figure 3 and Table 3 shows comparative analysis of performance metrics of the DMTCNN methods with other approaches. The graph establishes the improved specificity, sensitivity, f-measure and accuracy of the DL method. The specificity for the DNN, DCNN, ADD-EMRFODL, and Mask R-CNN models are 70.98%, 74.12%, 86.34%, and 70.12%, respectively, compared to the DMTCNN model's specificity of 94.18%. Similar, the recall for the DNN, DCNN, ADD-EMRFODL, and Mask R-CNN models are 81.78%, 70.77%, 67.23%, and 81.24%, respectively, compared to the DMTCNN model's recall of 93.19%. Like, the f-measure for the DNN, DCNN, ADD-EMRFODL, and Mask R-CNN models are 89.16%, 61.14%, 79.98%, and 79.18%, respectively, compared to the DMTCNN model's f-measure of 94.18%. Finally, the accuracy for the DNN, DCNN, ADD-EMRFODL, and Mask R-CNN models are 80.98%, 79.98%, 75.56%, and 71.23%, respectively, compared to the DMTCNN model's accuracy of 96.97%.

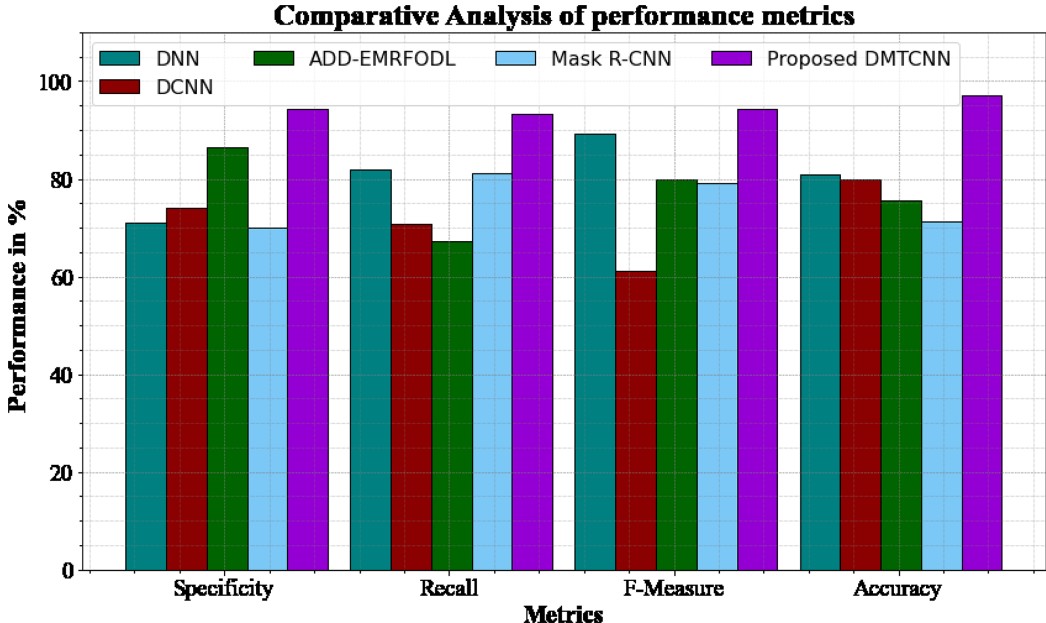

**Figure 3 Comparative analysis of performance metrics.**

**Table 3 Comparative analysis of performance metrics.**

| Methods | Specificity | Recall | F-measure | Accuracy |
|---|---|---|---|---|
| DNN | 70.98 | 81.78 | 89.16 | 80.98 |
| DCNN | 74.12 | 70.77 | 61.14 | 79.98 |
| ADD-EMRFODL | 86.34 | 67.23 | 79.98 | 75.56 |
| Mask R-CNN | 70.12 | 81.24 | 79.18 | 71.23 |
| DMCTNN | 94.18 | 93.19 | 94.18 | 96.97 |

### RMSE analysis

Figure 4 and Table 4 show an RMSE comparison of the DMTCNN strategy to other well-known methods. The graph demonstrates that the machine learning technique performs better with a lower RMSE. For example, the DMTCNN model has an RMSE value of 21.67% for 100 data, while the DNN, DCNN, ADD-EMRFODL, and Mask R-CNN models have RMSE values of 49.18%, 40.77%, 37.18%, and 28.19%, respectively. Moreover, the DMTCNN model has demonstrated superior performance for a wide range of data sizes with low RMSE values. Similarly, for 500 data, the RMSE score for the DMTCNN is 22.76%, whereas the DNN, DCNN, ADD-EMRFODL, and Mask R-CNN models are 46.11%, 41.87%, 38.16%, and 33.19%, respectively.

### Execution time analysis

The database execution time of the proposed DMTCNN approach is compared to known methods in Table 5 and Fig. 5. The statistics display that the proposed DMTCNN method outperformed all other approaches collectively. The suggested DMTCNN technique has

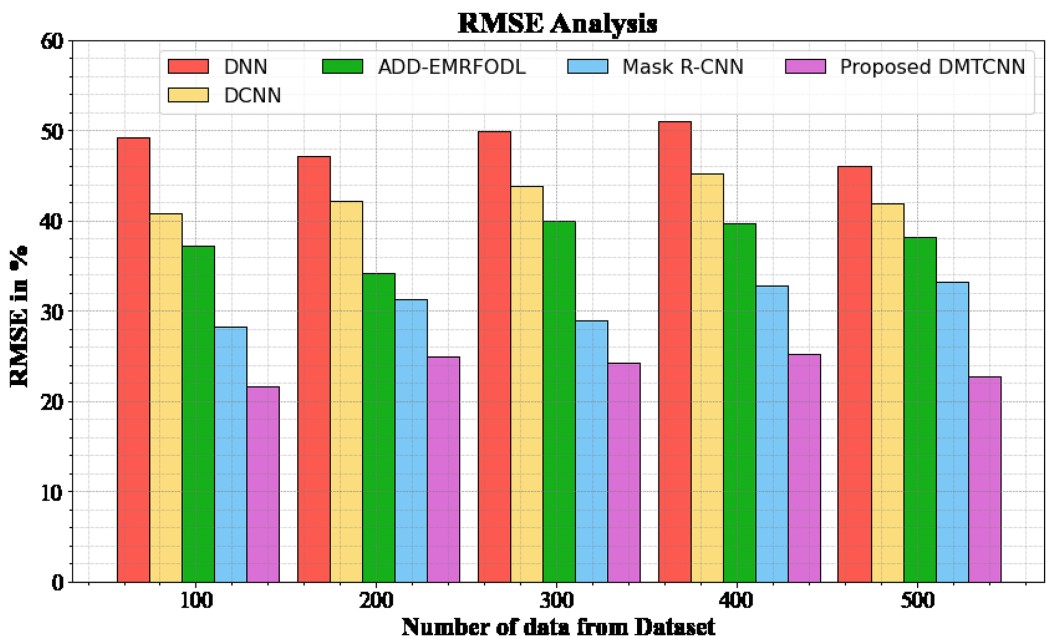

Figure 4 RMSE analysis for DMTCNN method.

Table 4 RMSE analysis for DMTCNN method.

| Number of data from dataset | DNN | DCNN | ADD-EMRFODL | Mask R-CNN | DMTCNN |
|---|---|---|---|---|---|
| 100 | 49.18 | 40.77 | 37.18 | 28.19 | 21.67 |
| 200 | 47.18 | 42.18 | 34.19 | 31.29 | 24.96 |
| 300 | 49.87 | 43.78 | 39.98 | 28.98 | 24.19 |
| 400 | 50.98 | 45.18 | 39.76 | 32.87 | 25.19 |
| 500 | 46.11 | 41.87 | 38.16 | 33.19 | 22.76 |

Table 5 Execution time analysis for DMTCNN technique.

| Number of data from dataset | DNN | DCNN | ADD-EMRFODL | Mask R-CNN | DMTCNN |
|---|---|---|---|---|---|
| 100 | 15.176 | 12.165 | 10.432 | 5.188 | 3.876 |
| 200 | 14.977 | 13.198 | 10.876 | 7.789 | 3.287 |
| 300 | 15.118 | 13.665 | 11.231 | 6.165 | 2.567 |
| 400 | 16.113 | 12.245 | 10.987 | 8.843 | 3.987 |
| 500 | 17.876 | 14.176 | 9.721 | 9.432 | 4.875 |

taken only 3.876 ms to execute, whereas other existing methods such as DNN, DCNN, ADD-EMRFODL, and Mask R-CNN took 15.176, 12.165, 10.432, and 5,188 ms, respectively, for 100 data their execution time. Likewise, the suggested DMTCNN technique executes in 4.875 ms for 500 data, but existing techniques such as DNN, DCNN, ADD-EMRFODL, and Mask R-CNN take 17.876, 14.176, 9.721, and 9.432 ms, correspondingly. In clinical contexts, where timely and accurate diagnoses are essential for

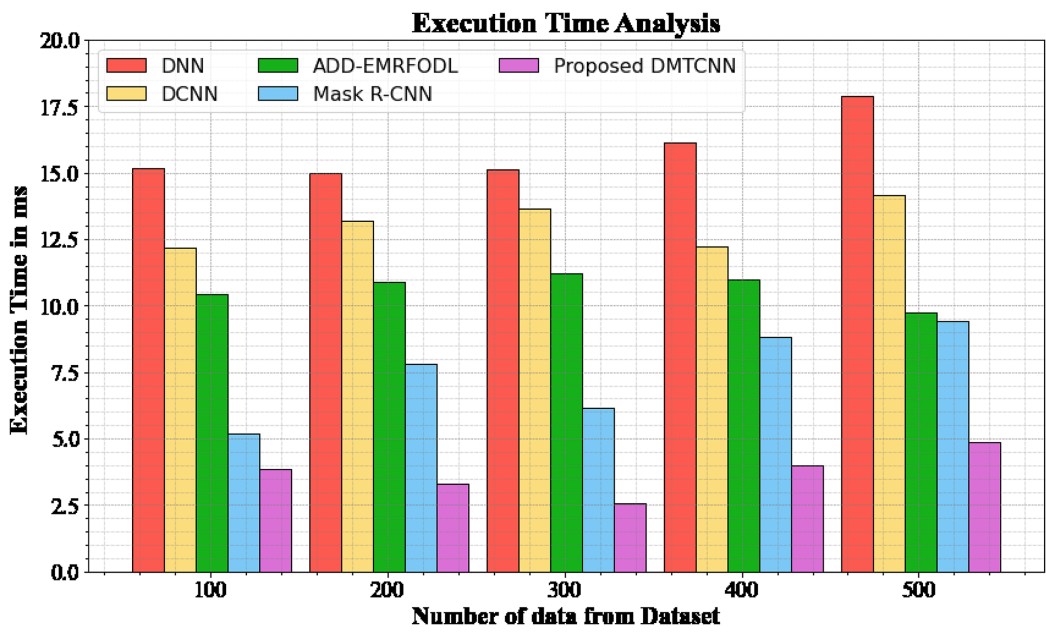

**Figure 5** Execution time analysis for DMTCNN technique.

patient care, this reduced execution time is particularly advantageous. In the context of swiftly progressing diseases such as brain disease diagnosis, faster inference not only enhances patient throughput but also enables more immediate interventions. DMTCNN is a practical and scalable solution for large-scale diagnostic applications due to its efficiency.

### Brain disease diagnosis analysis

Figure 6 shows the brain disease diagnosis analysis. In this work, we evaluated the suggested method on two different brain MRI datasets. Our main dataset, the Figshare brain tumor dataset, is one of the most comprehensive sets of data out there and is essential for the identification of brain tumors. 3,064 real brain MRI samples from 233 people are included in this collection. Of these, the meningioma class includes 708 samples, the pituitary class includes 930 examples, and the gliomas class includes 1,426 samples.

### Discussion

The improved efficacy of DMTCNN underscores the possible advantages of combining classification and regression assignments into a single, cohesive framework for diagnosing brain disorders. The network can classify more subtle patterns in the data and produce more reliable and thorough diagnostic insights by concurrently learning from many tasks. By concentrating the model's attention on relevant brain regions, the patch extraction technique suggestively progresses diagnostic accuracy. Following research endeavors ought to concentrate on the model's ability to scale and adapt to numerous clinical environments. To further ensure that the model's outputs are reliable and useful, investigating DMTCNN's interpretability may proposal useful data for clinical decision-making. The proposed DMTCNN framework, as established by experimental results, exhibits

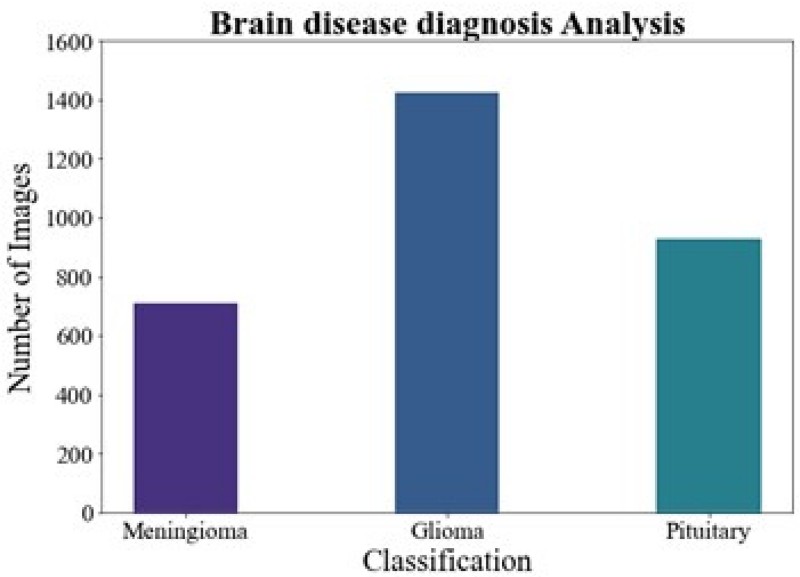

**Figure 6 Brain disease diagnosis analysis.**

exceptional performance with segmentation analysis accuracy at 96.97%, sensitivity at 93.19%, specificity at 94.18%, f-measure at 94.18%, RMSE at 22.76%, and an execution time of 4.875 ms.

## Ablation study

In the recommended model, every module is important. In this section, we inspect the framework for the proposed DMTCNN and associate it to recognized models such as DNN, DCNN, ADD-EMRFODL, and Mask R-CNN. This comparative study is carried out through a series of ablation studies using the Brain MRI Dataset to investigate the performance development and establish the motivations behind our proposed DMTCNN.

### Influence of the DMTCNN

The implementation of DMTCNN for patch extraction suggestively augments brain disease diagnosis by improving the accuracy and effectiveness of medical image analysis. DMTCNNs are capable of simultaneously handling multiple related tasks, such as segmentation and classification, which permits for a more comprehensive and nuanced understanding of brain abnormalities. By extracting and analyzing image patches, these networks can focus on relevant regions of interest, thereby growing diagnostic precision and decreasing the time necessary for manual inspection. This multi-task method not only streamlines the diagnostic procedure but also leads to better patient outcomes through previous and more accurate detection of brain diseases. Lastly, the DMTCNN model accomplished a performance accuracy of 96.97% for our input data. In contrast, the existing DNN, DCNN, ADD-EMRFODL, and Mask R-CNN models found accuracy performances of 80.98%, 79.98%, 75.56%, and 71.23% correspondingly.

| K-folds | DMTCNN accuracy |
| --- | --- |
| **Table 6** 10-fold cross validation of DMTCNN analysis. | |
| 1-Fold | 0.97 |
| 2-Fold | 0.96 |
| 3-Fold | 0.95 |
| 4-Fold | 0.96 |
| Fold-5 | 0.97 |
| Fold-6 | 0.98 |
| Fold-7 | 0.98 |
| Fold-8 | 0.97 |
| Fold-9 | 0.97 |
| Fold-10 | 0.97 |
| 10-Fold mean | 0.96 |

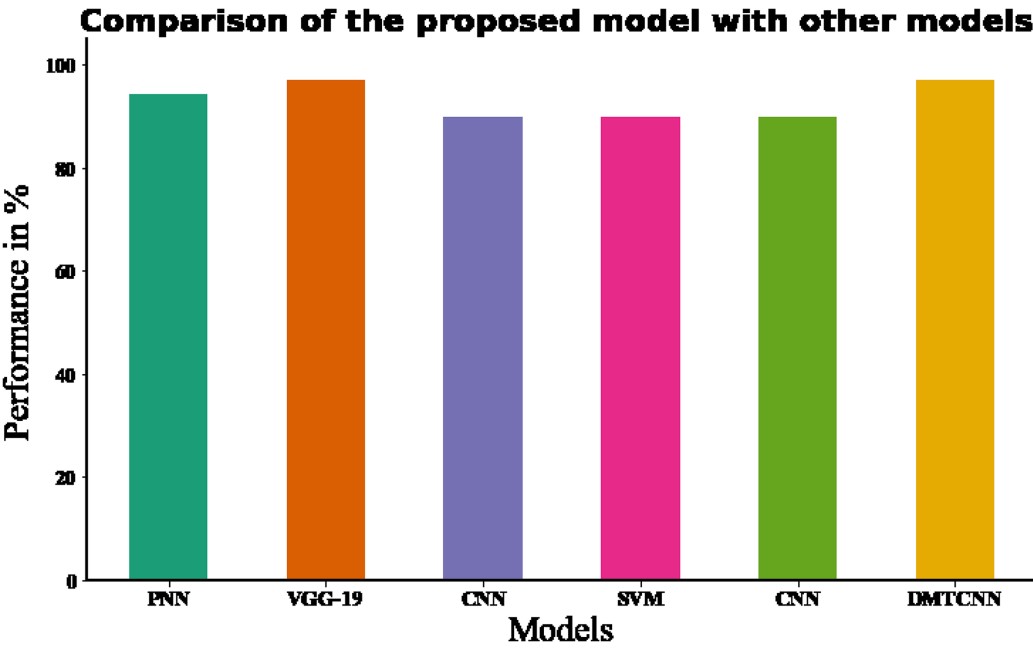

**Figure 7 Comparing the recommended model to state-of-the-art methodologies.**

### Influence of the K-fold cross validation

In DMTCNN-based brain disease diagnosis, the implementation of 10-fold cross-validation substantially improves model robustness and dependability. Ten subsets of the dataset are created as part of the statistical technique known as cross-validation. There are ten subsets: one is for training, and the other nine are for validation. The proposed DMTCNN model outperformed, obtaining 96.97% accuracy on our input data using 10-fold cross-validation. In comparison, the current DT, RF, KNN, and NB models have accuracy performances of 70.14%, 78.39%, 87.48%, and 85.26%, respectively (Table 6).

**Table 7 Comparison of the proposed model to current approaches (*Wang et al., 2024*).**

| Methods | Evaluation methods | Accuracy (%) | F-measure(%) |
|---|---|---|---|
| CNN (*Ahmed et al., 2020*) | 10-fold cross validation | 70.14 | 89.48 |
| SVM (*Collij et al., 2016*) | 10-fold cross validation | 78.39 | 90.34 |
| DT (*Vidushi & Shrivastava, 2019*) | 10-fold cross validation | 87.48 | 91.29 |
| RF (*Vidushi & Shrivastava, 2019*) | 10-fold cross validation | 85.26 | 92.34 |
| Proposed DMTCNN | 10-fold cross validation | 96.97 | 94.18 |

**Table 8 Comparison of the suggested model with innovative accuracy anlysis approaches.**

| Authors | Classifier | Database | Accuracy (%) | F-measure (%) |
|---|---|---|---|---|
| *Raghavendra et al. (2021)* | PNN (Probabilistic Neural Network) | Brain CT | 94.37% | – |
| *Ozaltin et al. (2022)* | VGG-19 | Brain CT | 97.06% | 87.43% |
| *Marbun & Andayani (2018)* | CNN | Brain CT | 90% | – |
| *Shalikar, Ashouri & Shahraki (2014)* | SVM | Brain CT | 90% | – |
| *Chin et al. (2017)* | CNN | Brain CT | 90% | – |
| Proposed model | DMTCNN | Brain CT | 96.97% | 94.18 |

Figure 7 shows the comparison of proposed method with existing model. Table 7 shows association of the proposed method to current approaches and Table 8 shows association of the suggested model with innovative accuracy analysis approaches.

## CONCLUSION

In conclusion, DMTCNN's combined classification and regression for brain disease detection through patch extraction may improve diagnostic accuracy and clinical decision-making. The purpose of the DMTCNN method to develop a more comprehensive and effective diagnostic tool that simultaneously classifies brain diseases and predicts the progression or severity of these conditions. This dual-task approach allows the model to provide richer insights, going beyond simple disease detection to also gauge the severity of neural degeneration or progression over time. The key objectives of the suggested model are to increase the interpretability of model predictions, increase the computational efficiency of traditional patch extraction methods that target meaningful and localized regions in brain images, and improve diagnostic accuracy by utilizing both classification and regression tasks in a single framework. Using a Canny edge detector for preprocessing and cooperative learning, DMTCNN is highly effective in predicting the existence of brain illness and quantitative metrics such as tumor volume. By optimizing numerous objectives, the model learns to extract significant features from neuroimaging data, including discriminative patterns for classification and quantitative associates for regression. The patch-based extraction technique also permits localized analysis of brain images to detect small anomalies that may not be visible globally. The efficiency of the proposed method DMTCNN was recognized by our experimental results, which showed notable gains in

important performance metrics with 500 data like specificity is 94.18%, sensitivity is 93.19%, accuracy is 96.97%, F1-score is 94.18%, RMSE is 22.76%, and execution time 4.875 ms. This fine-grained analysis progresses the time and accuracy of brain illness diagnosis and characterization. Deep learning in healthcare can be additional expanded by examining how this technique might be applied to numerous medical imaging modalities and kinds of illnesses in future studies. To provide a more entire diagnosis, the model might integrate multi-modal data such genetic data, patient history, and cognitive scores. Finally, healthcare uptake of the model will depend on its application to other neurological disorders, interpretability, and robustness in real-world clinical settings with varied populations.

### Funding
The authors received no funding for this work.

### Competing Interests
The authors declare that they have no competing interests.

### Author Contributions
- Padmapriya K. conceived and designed the experiments, analyzed the data, performed the computation work, prepared figures and/or tables, authored or reviewed drafts of the article, and approved the final draft.
- Ezhumalai Periyathambi performed the experiments, performed the computation work, prepared figures and/or tables, and approved the final draft.

### Data Availability
The data is available at figshare and Kaggle:

- Cheng, Jun (2017). brain tumor dataset. figshare. Dataset. https://doi.org/10.6084/m9.figshare.1512427.v5

- https://www.kaggle.com/navoneel/brain-mri-images-for-brain-tumor-detection.

### Supplemental Information
Supplemental information for this article can be found online at http://dx.doi.org/10.7717/peerj-cs.2538#supplemental-information.

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
