# Peer review of "Joint classification and regression with deep multi task learning model using conventional based patch extraction for brain disease diagnosis"

_PeerJ Computer Science, doi:10.7717/peerj-cs.2538_

## Round 0.1 · original submission · Major Revisions

Based on the referee reports, I recommend a major revision of the manuscript. The author should improve the manuscript, taking carefully into account the comments of the reviewers in the reports and resubmit the paper.

·

Basic reporting

Manuscript ID Submission ID 103324v1
This paper is related to reviewing the manuscript titled " Joint Classification and Regression with Deep multi task learning model using Conventional based patch extraction for brain disease diagnosis"
This work offers DMTCNN, a revolutionary deep learning model that can detect the existence of brain illnesses while also quantifying disease-related variables like tumor volume or disease severity. DMTCNN successfully recovers shared information from picture patches using multi-task learning, improving its generalization and diagnostic accuracy. The model includes a clever edge detector for pre-processing and use convolutional neural networks to extract useful characteristics. This novel technique shows great promise for improving the early diagnosis and measurement of brain disorders.

Experimental design

Firstly, Although the proposed study is successful in terms of organization, presentation, content and results, major revision given in the following items need to be performed.
1) Improve the conclusion section, enhance the manuscript to convey the purpose, objectives, method and major findings (In abstract section, too).
2) Use abbreviations after the first use in the text, in the abstract and throughout the paper.
3) Complete the introduction and literature sections of the article by providing more similar studies from the years 2023-2024 and/or new and current studies that will attract the attention of the readers.
4) It would be better if it were explained a little bit which of the brain MRI images in Figure 2 reflects what and what it means.
5) The x and y axes of all figures between Figure 4 and Figure 10 are not clear. Therefore, it is not understood what results the deep learning architectures compared produce. The same situation applies to Figure 12.
6) 1-Fold or Fold-1? Let this situation be made consistent.
7) Why is the reference in Table 9 (Reference 21)? Were the results in this table not obtained by the authors?
8) The resolution of the figures giving the analysis results should be increased.
9) Clean the paper of English spelling and punctuation errors

Validity of the findings

As above

Reviewer 2 ·

Basic reporting

- This manuscript presents a joint classification and regression with deep multi-task learning model using conventional based patch extraction for brain disease diagnosis.

- This manuscripts is well-written and easy to follow.

- Literatures are up-to-date.

- Visibility of tables and figures are good.

- Authors need to highlight in sub-section 2.1 after limitation of the existing systems.

Experimental design

- Highlight best results in bold in all tables.

- In table-9, compared your model with neural network baselines, such as CNN, LSTM, GRU, etc.

- Add a new table to describe hyper parameter values, like #neurons, optimiser, epoch, etc. for DMTCNN in sub-section 4.1

- Prepare one table only to show precision, recall, f-score, and accuracy instead of presenting it separately from tables 2 to 5. Also, mention the proposed model in bracket.

- In table 9, add F- score column

- In table 10, add F- score column

- Expand discussion section.

Validity of the findings

- Results are impressive as compared to the previous methods.

- Future works need to be expand more.

- Experimental results analysis should be expanded more.

Additional comments

This manuscripts need revision before acceptance.

Reviewer 3 ·

Basic reporting

- In the abstract, "this research" is misspelled in the method section. It should be changed to "In this research". Again, "an edge" should be used in the abstract, not "a edge". There are too many grammatical errors throughout the paper. The English of the paper should be reviewed by a native English speaker.
- The motivation in the "1.1 Motivation of this research" section is discussed. However, an explanation should be written after the title and then the motivation sentences given in bullet points should be moved on.
- Similarly, in section 1.2, an explanation should be written after the title.
- The studies discussed in the related studies section are analyzed in detail. However, this section can be expanded by considering more studies examining brain diseases, especially in recent years. Maybe a table can be added to this section and listed.
- The statements in the first sentence of Section 2.1 are very precise. Where was this information obtained? If the authors obtained such information themselves, how was the decision made? How was the mentioned sensitivity evaluated?
- Equation numbers should start from the Gaussian Smoothing section.
- Symbols should be written in italics (e.g.; f, G, and etc.)
- For each of the 4 steps in the Preprocessing section, the purpose for which it is used should be explained after the equations.
- Section 3.3. started with "However". Was there a statement before, it seems to be missing.
- Figure 2 and Table 1 are not referenced in the text.
- What do the axes represent in Figure 10?
- The long versions of the algorithm names DT, RF, K-NN, and NB should be given where they are first used. In addition, the information that these algorithms will be used should be added to the introduction section.

Experimental design

- In the introduction section, it is mentioned that a DMTCNN-based method is proposed for the diagnosis of brain diseases. In addition, dementia, brain tumors, and meningiomas are mentioned. Will the proposed method be used for the diagnosis of all diseases mentioned? This information should be presented clearly in the introduction section.
- Which tasks are meant by "several tasks" in the second sentence of the contributions section? It should be explained.

Validity of the findings

- What is the data splitting ratio?
- The conclusion section should be expanded.
- The parameter values ​​of the algorithms used should be added to the experimental results section with a table.
-The fact that the execution time for DMTCNN is so low should be commented on.

---

## Round 0.2 · accepted · Accept

The manuscript is ready for publication.

·

Basic reporting

The authors have made the requested corrections and taken the suggestions into consideration in the first round of revision. Therefore, I kindly request that this manuscript be accepted for publication in this journal.

Experimental design

None

Validity of the findings

None

Reviewer 2 ·

Basic reporting

Authors have addressed all comments

Experimental design

Authors have addressed all comments

Validity of the findings

Authors have addressed all comments

Additional comments

Accept

Reviewer 3 ·

Basic reporting

No comment

Experimental design

No comment

Validity of the findings

No comment

Additional comments

The authors have revised the article, taking into account all the necessary suggestions. I believe this paper, which has become more useful and understandable for the reader, can be accepted.